# Novel Approach to the Construction of Fused Indolizine Scaffolds: Synthesis of Rosettacin and the Aromathecin Family of Compounds

**DOI:** 10.3390/molecules28104059

**Published:** 2023-05-12

**Authors:** Shohta Mizuno, Takashi Nishiyama, Mai Endo, Koharu Sakoguchi, Takaki Yoshiura, Hana Bessho, Toshio Motoyashiki, Noriyuki Hatae, Tominari Choshi

**Affiliations:** 1Faculty of Pharmacy and Pharmaceutical Sciences, Fukuyama University, 1 Sanzo, Gakuen-cho, Fukuyama 729-0292, Japan; p7118100@fukuyama-u.ac.jp (S.M.); t_nishiyama@fukuyama-u.ac.jp (T.N.); ma.tetubas@gmail.com (M.E.); p7119044@fukuyama-u.ac.jp (K.S.); p7119105@fukuyama-u.ac.jp (T.Y.); p71120085@fukuyama-u.ac.jp (H.B.); motoyashiki@fukuyama-u.ac.jp (T.M.); 2Faculty of Pharmaceutical Sciences, Yokohama University of Pharmacy, 601 Matano, Totsuka-ku, Yokohama 245-0066, Japan; noriyuki.hatae@hamayaku.ac.jp

**Keywords:** acuminatine, 22-hydroxyacuminatine, benz[6,7]indolizino[1,2-*b*]quinolin-11(13*H*)-one, total synthesis, thermal cyclization

## Abstract

Camptothecin-like compounds are actively employed as anticancer drugs in clinical treatments. The aromathecin family of compounds, which contains the same indazolidine core structure as the camptothecin family of compounds, is also expected to display promising anticancer activity. Therefore, the development of a suitable and scalable synthetic method of aromathecin synthesis is of great research interest. In this study, we report the development of a new synthetic approach for constructing the pentacyclic scaffold of the aromathecin family by forming the indolizidine moiety after synthesizing the isoquinolone moiety. Thermal cyclization of 2-alkynylbenzaldehyde oxime to the isoquinoline *N*-oxide, followed by a Reissert–Henze-type reaction, forms the key strategy in this isoquinolone synthesis. Under the optimum reaction conditions for the Reissert–Henze-type reaction step, microwave irradiation-assisted heating of the purified *N*-oxide in acetic anhydride at 50 °C reduced the formation of the 4-acetoxyisoquinoline byproduct to deliver the desired isoquinolone at a 73% yield after just 3.5 h. The eight-step sequence employed afforded rosettacin (simplest member of the aromathecin family) at a 23.8% overall yield. The synthesis of rosettacin analogs was achieved by applying the developed strategy and may be generally applicable to the production of other fused indolizidine compounds.

## 1. Introduction

The indolizidine and quinolizidine moieties are important core structures that can be found in several biologically active compounds, such as camptothecin (**1**) [1], rosettacin (**5**) [2], and the 8-oxoprotoberberine alkaloids **8**–**10** [3,4] (Figure 1). The study of indolizidine- or quinolizidine-containing compounds has attracted the interest of many research groups because these compounds exhibit antitumor activity. Among such compounds, camptothecin (**1**) was isolated by Wani and co-workers in 1966 from the Chinese tree *Camptotheca acuminata* [1]. This alkaloid was shown to potently inhibit tumor growth by binding to the topoisomerase I enzyme (Top1). Subsequently, drug development studies were conducted with **1** as the lead compound. As a result, irinotecan (**2**) [5], topotecan (**3**) [6], and belotecan (**4**) [7], all compounds bearing substituents on the AB-ring, were developed for use in clinical studies. However, hydrolysis of the E-ring lactone moiety produces hydroxycarboxylates with a high affinity for human serum albumin protein; thus, the E-ring lactone hydrolysis product is responsible for attenuating the activity of **1** derivatives [8]. To address this drawback, the development novel anticancer drugs has focused on the aromathecin family of compounds (benz[6,7]indolizino[1,2-*b*]quinolin-11(13*H*)-ones), in which the lactone moiety of **1** is replaced with a benzene ring.

To date, three members of the aromathecin family are known: rosettacin (**5**), 22-hydroxyacuminatine (**6**) [6], and acuminatine (**7**) [7]. Together with **1**, **6** has been isolated at a very low yield from the seeds of *C. accuminata* [6]. Moreover, rosettacin, its derivatives, and **6** have been reported to display weak Top1 inhibitory activity [11,12,13,14,15,16]. Therefore, aromathecins can be considered as a new class of Top1 inhibitors which can replace camptothecins as candidates for therapeutic development. To date, many approaches have been reported for the synthesis of aromathecins. These routes primarily focus on a late-stage indolizidine moiety construction by either C- or D-ring formation [15,17,18,19,20,21,22,23,24,25,26]. Indeed, several synthetic procedures have been developed in recent years which enable the facile production of aromathecins. Eycken and co-workers reported the total synthesis of **5** via three synthetic strategies [17,18,19]. The first approach is based on using an intramolecular cascade annulation of *O*-substituted *N*-hydroxybenzamides, triggered by Rh(III)-catalyzed sequential C(sp^2^)-H activation and C(sp^3^)-H amination, for the synthesis of indolizinones [17]. The next method involves the intermolecular annulation of 2-acetylenic aldehydes with *O*-substituted *N*-hydroxybenzamides through Rh(III)-catalyzed C–H activation for the synthesis of indolizinones [18]. The third route comprised the construction of indolizinones by an intramolecular cascade annulation triggered by C–H activation via rhodium hydride intermediate [19]. Huang and co-workers reported the synthesis of **5** via the construction of isoquinolone using carbene-catalyzed aerobic oxidation of isoquinolinium salts, followed by a Pd-catalyzed intramolecular cyclization [20]. Reddy and co-workers developed the synthesis of **5** by a one-pot method for the synthesis of 7-hydroxyisoindolo[2,1-*b*]isoquinolin-5(7*H*)-ones from *N*-(pivaloyloxy)benzamides and 2-alkynyl aldehydes via Rh(III)-catalyzed C–H functionalization [21]. Furthermore, Evano and co-workers developed a method to assemble the indolizinone moiety by copper-catalyzed photoinduced radical domino cyclization of ynamides and applied this method to the synthesis of **5** [22]. Glorius and co-workers reported the synthesis of **5** and its derivatives via the construction of isoquinolones by the intramolecular annulation of a Cp*Co^III^-catalyzed C–H activation approach for *N*-(pent-4-yn-1-yloxy)benzamide production as the key step [23].

Our research group has been interested in the unique structure and pharmacological action of condensed heteroaromatic compounds and we have been searching for highly active compounds based on these naturally occurring compounds and their derivatives. To date, we have achieved the total synthesis of several such compounds using various types of 6π-electron system (hexatriene, 1- and 2-azahexatriene) electrocyclic reactions, such as asiaticumine [27], marinoquinolines [28], trigonoine B [29], girinimbine [30], and karnatakafuran B [31]. We have also published the total synthesis of (*R*)-(–)-pyridindolols using another synthetic approach that involved thermal cyclization of 3-alkynylindole-2-aldoxime to construct the key β-carboline *N*-oxide intermediate [32,33].

Recently, we developed a versatile synthetic route to produce 8-oxoprotoberberine alkaloids (alangiumkaloids A (**9**) and B (**10**)) through the synthesis of isoquinolone and via B-ring construction [34] (Figure 1). The key step in this was the synthesis of isoquinolone **13** through the thermal cyclization of 2-alkynylbenzaldehyde oxime **11** to afford isoquinoline *N*-oxide **12**, followed by a Reissert–Henze-type reaction. The lithium aluminum hydride (LiAlH_4_) reduction of both the obtained isoquinolone **13** and 1-acetoxyisoquinoline **14**, followed by quinolizidine ring construction (B-ring formation), completed the first total synthesis of alangiumkaloids **9** and **10**. Employing this strategy, we also synthesized a series of alangiumkaloid analogs and are currently conducting exploratory antitumor activity research into these compounds.

In this paper, we report on the synthesis of rosettacin (**5**) and its derivatives, a feat achieved by applying the synthetic strategies from our previous work to construction of a benz[6,7]indolizino[1,2-*b*]quinolin-11(13*H*)-one scaffold.

## 2. Results and Discussion

Our retrosynthetic analysis of **5** is presented in Figure 2. The rosettacin scaffold can be synthesized by late-stage construction of the indolizine ring (CD-ring) moiety through bond formation between C13 and N12. The precursor isoquinolone **15** (DE-ring) can be obtained from a Reissert–Henze-type reaction of isoquinoline *N*-oxide **16**, which can be formed from the thermal cyclization of 2-alkynylbenzaldehyde oxime **17**. We envisioned that **17** could be synthesized from the Sonogashira reaction between 2-iodoquinoline **18** and 2-ethynylbenzaldehyde **19**. The advantage of this synthetic strategy was that it allowed for easy access to rosettacin analogs by derivatizing both the starting quinolines and 2-ethynylbenzaldehydes.

We began the synthesis of three quinolines (AB-ring) in possession of the necessary substitutions, as shown in Figure 3. The starting materials, 2-chloroquinolines **18a** and **18b**, were synthesized from acetanilide according to the method of Bhuyan and co-workers [35]. Subsequently, 2-chloroquinolines **18a** and **18b** were heated with NaI and concentrated HCl in acetonitrile (MeCN) to obtain 2-iodoquinolines **20a** and **20b** at 92% and 90% yields, respectively [36]. Treatment of **20b** with iodine and K_2_CO_3_ in methanol (MeOH) provided methyl ester **21** at an 89% yield. Alternatively, the reduction of **20b** with NaBH_4_ followed by hydroxy group methylation provided 3-methoxymethylquinoline **23** at an 80% yield over 2 steps.

Next, we prepared the key isoquinolone synthesis precursors: 2-alkynylbenzaldehyde oximes **25** (Figure 4). The Sonogashira reaction of 2-iodoquinolines **21**, **20a**, and **23** with 2-ethynylbenzaldehyde **19** in the presence of CuI, triethylamine (Et_3_N) and bis(triphenylphosphine)palladium(II) chloride (PdCl_2_(PPh_3_)_2_) provided 2-alkynylbenzaldehydes **24a**, **24b**, and **24c** at 44%, 71%, and 84% yields, respectively. The treatment of **24a**–**c** with hydroxylamine produced oximes **25a**–**c** at 68%, 53%, and 86% yields, respectively.

As shown in Table 1, we next investigated the synthesis of isoquinolone **27** via the construction of isoquinoline *N*-oxide **26** from oxime **25**. First, oxime **25a** was heated in 1,2-dichlorobenzene (1,2-DCB) at 180 °C until the starting material was no longer detectable by thin-layer chromatography (TLC) analysis. The reaction was quenched by the evaporation of 1,2-DCB in vacuo to obtain *N*-oxide **26**. Subsequently, without further purification, the crude *N*-oxide **26a** was heated in acetic anhydride (Ac_2_O) at 110 °C. Disappointingly, the desired isoquinolone **27a** was not obtained and an unidentifiable mixture of compounds was produced (entry 1). Reducing the cyclization temperature to 80 °C, the thermal cyclization of **25a** proceeded smoothly; however, the subsequent heating of the resulting crude **26a** in Ac_2_O at 110 °C also did not afford isoquinoline **27a** (entry 2). Therefore, compounds containing an ester functional group were deemed to be unsuitable for the Reissert–Henze-type reaction and studies of oxime **25a** ceased.

Next, we investigated the synthesis of isoquinolone **27b** from oxime **25b**. *N*-oxide **26b** was obtained by heating oxime **25b** in 1,2-DCB at 80 °C. Thereafter, crude **26b** was heated in Ac_2_O at 110 °C to afford the desired **27b** along with acetoxy **28b** at 6% and 33% yields, respectively (entry 3).

Subsequently, the synthesis of isoquinolone **27c** from oxime **25c** was investigated according to Method A (entry 4). Isoquinolone **27c** and the acetoxy product **28c** were obtained at 22% and 40% yields, respectively. From these exploratory results, oxime **25c** was selected for use in further optimization studies of the Reissert–Henze-type reaction to isoquinolone **27c**.

To examine the effect of *N*-oxide **26c** purity on the subsequent reaction, the crude *N*-oxide **26c** obtained by thermal cyclization was purified by crystallization (purification yield 73%). Purified *N*-oxide **26c** was heated in Ac_2_O at 110 °C (entry 5), resulting in slightly improved yields of **27c** (from 22% to 31%). When employing crude **26c**, lowering the Reissert–Henze-type reaction temperature to 50 °C further improved the yield of **27c** to 48% yield (entry 6). Notably, the production of acetoxy **28c** was reduced at the lower reaction temperature. Returning to Method B, the reaction of purified **26c** at 50 °C did not improve the yield of **27c** (entry 7). However, employing ambient temperature (rt) conditions, stirring purified **26c** in Ac_2_O improved the yield of **27c** to 60% and further reduced the production of **28c** to 18% (entry 8), although the reaction time required was significantly longer (51 h). From these early optimization studies, it may be asserted that the Reissert–Henze-type synthesis of isoquinolone **27c** produced better yields when starting from the purified *N*-oxide **26c**.

Next, the effect of microwave (MW) technology on the Reissert–Henze-type reaction was studied. Unfortunately, the yield of **27c** decreased to 20% when purified **26c** was heated in Ac_2_O at 110 °C under MW irradiation (entry 9). To directly compare the effects of MW irradiation (entry 6 vs. entry 10), crude **26c** was heated in Ac_2_O at 50 °C under MW, the reaction time required was greatly reduced from 24 h to 14 h and the production of **28c** was further reduced. Gratifyingly, using the same conditions of entry 10, purified **26c** was transformed into **27c** with a greatly improved yield of 73% in a short 3.5 h reaction time. Briefly, the use of trifluoroacetic anhydride in place of Ac_2_O and the addition of an acid catalyst were investigated; however, the yield of **27c** did not improve further (entries 12 and 13). Therefore, the conditions of entry 11 were deemed optimal for the Reissert–Henze-type synthesis of isoquinolone **27c** from **25c**.

During our Reissert–Henze-type reaction optimization studies, because of our work on the synthesis of alangiumkaloids (Figure 1), we predicted that the obtained acetoxy **28c** was the 1-acetoxyisoquinoline product (Figure 5). Therefore, we attempted to remove the acetyl group of **28c** by LiAlH_4_ reduction. However, the ^1^H-NMR spectra of the obtained compound **29** did not match that of isoquinolone **27c**. Further 2D-NMR studies (heteronuclear multiple-bond correlation (HMBC) and nuclear Overhauser effect spectroscopy (NOESY) measurements) were performed to verify the structure of the obtained **28c**. The through-space proton interaction correlations are shown in Figure 2. In particular, **28c** was identified as 4-acetoxyisoquinoline because the NOESY correlation of the methyl protons (2.13 ppm) of the acetyl group to C5-H (7.98 ppm) and C8’-H (8.13 ppm) was observed. Consequently, compound **29**, obtained from **28c** reduction, was confirmed to be 4-hydroxy-3-(3-methoxymethylquinolin-2-yl)isoquinoline and compound **28b** was also confirmed to be 4-acetoxyisoquinoline. The formation of both the 4-acetoxyisoquinoline product together with the isoquinolone product has been reported by Robison and co-workers [37]. However, why the Reissert–Henze-type reaction of our *N*-oxide **26** afforded 4-acetoxyisoquinoline remains to be elucidated. In the future, we will investigate this phenomenon in greater detail.

To complete the synthesis of rosettacin (**5**), we applied the conditions reported by Ciufolini and co-workers for C-ring formation [38]. Heating **27c** with H_2_SO_4_ in ethanol (EtOH) produced **5** at an 88% yield (Figure 6). Thus, the total synthesis of rosettacin (**5**) was achieved through an eight-step sequence at a 23.8% overall yield.

As described above, we established a method for the synthesis of benz[6,7]indolizino[1,2-*b*]quinolin-11(13*H*)-one, which is the core structure of compounds in the aromathecin family. Next, we tried to synthesize 22-hydroxyacuminatin (**6**) by applying the same methodology (Figure 7). Sonogashira coupling of 2-iodoquinoline **23** and 2-ethynylbenzaldehyde **30** in the presence of CuI, Et_3_N and PdCl_2_(PPh_3_)_2_ produced 2-alkynylbenzaldehyde **31** at an 87% yield. The treatment of **31** with hydroxylamine in EtOH produced 2-alkynylbenzaldehyde oxime **32** at an 81% yield. Subsequently, heating **32** in 1,2-DCB at 80 °C did not deliver the isoquinoline *N*-oxide cyclization product **33**. Instead, the treatment of **32** in 1,2-DCB at 180 °C produced **33** at a 36% yield. Next, heating **33** in Ac_2_O at 50 °C under MW irradiation produced the desired isoquinolone **34** and 4-acetoxyisoquinoline **35** at yields of 54% and 20%, respectively. Finally, when **34** was heated with H_2_SO_4_ in EtOH, the desired 22-hydroxyacuminatine (**6**) was not produced, but acuminatine (**7**) was isolated at a 79% yield. The formation of **7** is believed to have proceeded with dehydroxylation after the removal of the MOM group along with scaffold formation. The physical and spectroscopic data for our synthetic rosettacin (**5**) were consistent with previously reported values in all respects [26]. Furthermore, ^1^H-NMR, ^13^C-NMR, and mass spectroscopy characterization of all our synthetic compounds supported the identified structures, the details of which can be found in the Appendix A section.

## 3. Materials and Methods

All non-aqueous reactions were carried out under an atmosphere of nitrogen in dried glassware unless otherwise noted. Solvents were dried and distilled according to standard protocols. Analytical thin-layer chromatography was performed with silica gel 60PF_254_ (Merck). Silica gel column chromatography was performed with silica gel 60 (70–230 mesh, Kanto Chemical Co. Lit., Nihonbashi, Tokyo, Japan). All melting points were determined on Yanagimoto micromelting point apparatus MP-500D (Yanaco Technical Sciences Co. Lit., Taito-ku, Tokyo, Japan) and are uncorrected. Proton nuclear magnetic resonance (^1^H-NMR) spectra were recorded on a JEOL JNM-ECZ400S (JEOL Resonance Co. Lit., Akishima, Tokyo, Japan). Chemical shifts were reported relative to Me_4_Si (δ 0.00). Multiplicity is indicated by one or more of the following: s (singlet); d (doublet); t (triplet); q (quartet); m (multiplet); br (broad). Carbon nuclear magnetic resonance (^13^C-NMR) spectra were recorded on a JEOL JNM-ECZ400S z at 100 MHz. Chemical shifts were reported relative to CDCl_3_ (δ 77.0) and DMSO-*d_6_* (δ 39.7). Infrared spectra were recorded with ATR method using a Horiba FT-720 FREEXACT-II spectrophotometer (Horiba Ltd., Kyoto, Japan) and Technologies DuraScop (ST. Japan Inc., Chuo-ku, Tokyo, Japan). Low- and high-resolution mass spectra were recorded on JEOL JMS-700 spectrometers (JEOL Resonance Co.Lit., Akishima, Tokyo, Japan) via the use of a direct inlet system. The microwave-assisted reaction was carried out at 180 W and 2450 MHz with Discover (CEM corporation, Matthews NC, USA).

### 3.1. 2-Iodo-3-methylquinoline (***20a***)

We added dropwise conc. HCl (0.12 mL) to a solution of 2-chloroquinoline **18a** (685 mg, 3.86 mol) and NaI (1.74 × 10^3^ mg, 11.61 mmol) in MeCN (16 mL); then, these were stirred at 85 °C for 16 h. After cooling at ambient temperature, we added H_2_O (15 mL) and saturated Na_2_S_2_O_3_ (15 mL) to the reaction mixture. The resulting precipitate was filtrated in vacuo. The filtrate was extracted with EtOAc. The organic layer was washed with H_2_O and brine, dried with Na_2_SO_4_, and evaporated in vacuo. The residue was purified by column chromatography (EtOAc/hexane 1:4 *v*/*v*) to give 2-iodoquinoline **20a** (958 mg, 92%) as a yellow solid. mp 181–183 °C (EtOAc-hexane). ^1^H-NMR (400 MHz, DMSO-*d_6_*) δ 2.46 (s, 3H), 7.61 (t, *J* = 8.2 Hz, 1H), 7.71 (t, *J* = 8.2 Hz, 1H), 7.89–7.93 (m, 2H), 8.17 (s, 1H). ^13^C-NMR (100 MHz, DMSO-*d_6_*) δ 25.8, 127.2, 127.3, 127.5, 127.6, 128.2, 129.6, 135.11, 135.14, 147.1. MS *m*/*z*: 269 (M^+^). HRMS (EI): calcd for C_10_H_8_NI 268.9701; found 268.9711.

### 3.2. 2-Iodoquinoline-3-carbaldehyde (***20b***)

The same procedure as above was carried out with 2-chloroquinoline **18b** (1.0 × 10^4^ mg, 52.35 mmol) to give 2-iodoquinoline **20b** (13.3 g, 90%) as a white solid. mp 150–151 °C (EtOH). IR (ATR) ν = 1685 cm^−1^. ^1^H-NMR (400 MHz, CDCl_3_) δ 7.67 (t, *J* = 8.2 Hz, 1H), 7.88 (t, *J* = 8.2 Hz, 1H), 7.97 (d, *J* = 8.2 Hz, 1H), 8.11 (d, *J* = 8.2 Hz, 1H), 8.56 (s, 1H), 10.29 (s, 1H). ^13^C-NMR (100 MHz, CDCl_3_) δ 120.5, 126.5, 128.3, 128.7, 128.9, 129.7, 133.4, 138.7, 151.6, 194.7. MS *m*/*z*: 283 (M^+^). HRMS (EI): calcd for C_10_H_6_NOI 282.9494; found 282.9485.

### 3.3. Methyl 2-Iodoquinoline-3-carboxylate (***21***)

A suspension of quinoline-3-carbaldehyde **20b** (200 mg 0.71 mmol), I_2_ (683 mg, 2.69 mmol) and K_2_CO_3_ (345 mg, 2.50 mmol) in MeOH (6 mL) was stirred at rt for 35 min. After quenching with H_2_O (5 mL) and saturated Na_2_S_2_O_3_ (5 mL), the resulting precipitate was filtered in vacuo to give methyl ester **21** (198 mg, 89%) as a yellow solid. mp 93–94 °C (EtOH). IR (ATR) ν = 1716 cm^−1^. ^1^H-NMR (400 MHz, CDCl_3_) δ 4.02 (s, 3H), 7.64 (t, *J* = 8.2 Hz, 1H), 7.82 (t, *J* = 8.2 Hz, 1H), 7.87 (d, *J* = 8.2 Hz, 1H), 8.10 (d, *J* = 8.2 Hz, 1H), 8.49 (s, 1H). ^13^C-NMR (100 MHz, CDCl_3_) δ 52.9, 115.7, 125.6, 128.0, 128.5, 128.8, 129.0, 132.3, 138.9, 149.8, 166.0. MS *m*/*z*: 313 (M^+^). HRMS (EI): calcd for C_11_H_8_NO_2_I 312.9600; found 312.9605.

### 3.4. 2-Iodoquinolin-3-ylmethanol (***22***)

A solution of quinoline-3-carbaldehyde **20b** (900 mg, 3.18 mmol) in MeOH (20 mL) was added dropwise to a suspension of NaBH_4_ (182 mg, 4.77 mmol) in MeOH (10 mL) under ice cooling conditions. After stirring at rt for 1 h, the reaction mixture was quenched with H_2_O. The resulting precipitate was filtrated in vacuo to give alcohol **22** (848 mg, 93%) as a yellow solid. mp 177–179 °C (CHCl_3_). IR (ATR) ν = 3745 cm^−1^. ^1^H-NMR (400 MHz, DMSO-*d_6_*) δ 4.53 (d, *J* = 8.2 Hz, 2H), 5.74 (t, *J* = 8.2 Hz, 1H), 7.64 (t, *J* = 8.2 Hz, 1H), 7.74 (t, *J* = 8.2 Hz, 1H), 7.95 (d, *J* = 8.2 Hz, 1H), 8.04 (d, *J* = 8.2 Hz, 1H), 8.22 (s, 1H). ^13^C-NMR (100 MHz, DMSO-*d_6_*) δ 65.5, 123.1, 127.1, 127.3, 127.6, 128.0, 129.9, 133.5, 137.7, 147.9. MS *m*/*z*: 285 (M^+^). HRMS (EI): calcd for C_10_H_8_NOI 284.9651; found 284.9663.

### 3.5. 2-Iodo-3-methoxymethylquinoline (***23***)

A solution of alcohol **22** (600 mg, 2.11 mmol) in THF (10 mL) was added dropwise to a suspension of 60% NaH (336 mg, 8.42 mmol) in THF (5 mL) under ice cooling. After stirring at the same temperature for 15 min, MeI (710 mg, 5.00 mmol) was added to the reaction mixture and this was then stirred at rt for 1 h. After quenching with H_2_O, the reaction mixture was extracted with EtOAc. The organic layer was washed with brine, dried with Na_2_SO_4_, and evaporated in vacuo. The residue was purified by column chromatography (EtOAc/hexane 1:9 *v*/*v*) to produce 3-methoxymethylquinoline **23** (543 mg, 86%) as a yellow solid. mp 45–46 °C (EtOAc-hexane). ^1^H-NMR (400 MHz, CDCl_3_) δ 3.58 (s, 3H), 4.55 (s, 2H), 7.57 (t, *J* = 8.1 Hz, 1H), 7.70 (t, *J* = 8.1 Hz, 1H), 7.82 (d, *J* = 8.1 Hz, 1H), 8.04–8.06 (m, 2H). ^13^C-NMR (100 MHz, CDCl_3_) δ 58.9, 76.4, 122.2, 127.3, 127.6, 128.4, 129.9, 134.2, 134.3 (2C), 148.7. MS *m*/*z*: 299 (M^+^). HRMS (EI): calcd for C_11_H_10_NOI 298.9807; found 298.9807.

### 3.6. Methyl 2-(2-Formylphenyl)ethynylquinoline-3-carboxylate (***24a***)

We added a solution of 2-ethynylbenzaldehyde **19** (62 mg, 0.48 mmol) in THF (1 mL) to a solution of 2-iodoquinoline **21** (100 mg, 0.32 mmol), CuI (6 mg, 0.032 mmol), PdCl_2_(PPh_3_)_2_ (7 mg, 0.0096 mmol) and Et_3_N (2 mL, 14.47 mmol) in THF (5 mL). The reaction mixture was stirred at 60 °C for 1 h. After cooling to ambient temperature, the reaction mixture was filtrated through Celite pad, washed with EtOAc, and the filtrate was evaporated in vacuo. The residue was purified using column chromatography (EtOAc/hexane 1:4 *v*/*v*) to give 2-alkynylbenzaldehyde **24a** (43 mg, 44%) as a yellow solid. mp 131–132 °C (EtOAc-hexane). IR (ATR) ν = 1689, 1709 cm^−1^. ^1^H-NMR (400 MHz, CDCl_3_) δ 4.03 (s, 3H), 7.53 (t, *J* = 7.8 Hz, 1H), 7.62–7.68 (m, 2H), 7.85–7.89 (m, 2H), 7.93 (d, *J* = 7.8 Hz, 1H), 8.01 (d, i = 7.8 Hz, 1H), 8.19 (d, *J* = 7.8 Hz, 1H), 8.87 (s, 1H), 10.86 (s, 1H). ^13^C-NMR (100 MHz, CDCl_3_) δ 52.8, 88.5, 94.6, 125.1, 125.9, 126.0, 127.0, 128.4, 128.6, 129.3, 129.6, 132.5, 133.7, 134.0, 136.9, 140.0, 141.0, 149.1, 165.2, 192.4. MS *m*/*z*: 315 (M^+^). HRMS (EI): calcd for C_20_H_13_NO 315.0895; found 315.0887.

### 3.7. 2-(3-Methylquinolin-2-yl)ethynylbenzaldehyde (***24b***)

The same procedure as above was carried out with 2-iodoquinoline **20a** (700 mg, 2.60 mmol) to give 2-alkynylbenzaldehyde **24b** (500 mg, 71%) as a white solid. mp 129–131 °C (EtOAc-hexane). IR (ATR) ν = 1689 cm^−1^. ^1^H-NMR (400 MHz, CDCl_3_) δ 2.71 (s, 3H), 7.52–7.57 (m, 2H), 7.64 (t, *J* = 7.8 Hz, 1H), 7.69 (t, *J* = 7.8 Hz, 1H), 7.76 (d, *J* = 7.8 Hz, 1H), 7.83 (d, *J* = 7.8 Hz, 1H), 7.99–8.02 (m, 2H), 8.11 (d, *J* = 7.8 Hz, 1H), 10.75 (s, 1H). ^13^C-NMR (100 MHz, CDCl_3_) δ 19.8, 88.2, 94.3, 125.5, 126.8, 127.4, 127.7 (2C), 128.9, 129.2, 129.4, 132.5, 133.7, 134.0, 135.5, 136.2, 143.6, 146.7, 191.1. MS *m*/*z*: 271 (M^+^). HRMS (EI): calcd for C_19_H_13_NO 271.0997; found 271.0988.

### 3.8. 2-(3-Methoxymethylquinolin-2-yl)ethynylbenzaldehyde (***24c***)

The same procedure as above was carried out with 2-iodoquinoline **23** (612 mg, 2.05 mmol) to give 2-alkynylbenzaldehyde **24c** (519 mg, 84%) as a white solid. mp 97–98 °C (EtOAc-hexane). IR (ATR) ν = 1693 cm^−1^. ^1^H-NMR (400 MHz, CDCl_3_) δ 3.57 (s, 3H), 4.87 (s, 2H), 7.52–7.61 (m, 2H), 7.65 (t, *J* = 7.8 Hz, 1H), 7.74 (t, *J* = 7.8 Hz, 1H), 7.76–7.82 (m, 2H), 8.01 (t, *J* = 7.8 Hz, 1H), 8.14 (t, *J* = 8.6 Hz, 1H), 8.27 (s, 1H), 10.72 (s, 1H). ^13^C-NMR (100 MHz, CDCl_3_) δ 59.0, 71.7, 88.6, 93.2, 125.3, 127.5, 127.6, 127.7, 127.8, 129.1, 129.6, 130.0, 132.9, 133.8, 134.1, 134.6, 136.5, 141.7, 147.6, 191.3. MS *m*/*z*: 301 (M^+^). HRMS (EI): calcd for C_20_H_15_NO_2_ 301.1103; found 301.1121.

### 3.9. Methyl 2-(2-Hydroxyiminomethylphenyl)ethynylquinoline-3-carboxylate (***25a***)

A mixture of benzaldehyde **24a** (70 mg, 0.22 mmol), NH_2_OH·HCl (30 mg, 0.44 mmol), and AcONa (36 mg, 0.44 mmol) in EtOH (5 mL) was stirred at rt for 1 h. After the removal of the solvent, the residue was diluted with H_2_O and then filtrated off to give crude oxime **25a** (50 mg, 68%) as a yellow solid. The product was recrystallized from EtOAc-hexane. mp 171–173 °C (EtOAc-hexane). IR (ATR) ν = 3047, 1709 cm^−1^. ^1^H-NMR (400 MHz, DMSO-*d_6_*) δ 3.99 (s, 3H), 7.43–7.52 (m, 2H), 7.71–7.77 (m, 2H), 7.91–7.97 (m, 2H), 8.11 (d, *J* = 7.8 Hz, 1H), 8.20 (d, *J* = 7.8 Hz, 1H), 8.73 (s, 1H), 9.03 (s, 1H), 11.67 (s, 1H). ^13^C-NMR (100 MHz, DMSO-*d_6_*) δ 53.3, 89.6, 93.8, 121.1, 125.1, 125.7, 126.2, 128.9, 129.0, 129.7, 130.1, 130.7, 133.4, 133.6, 135.3, 140.2, 140.6, 146.6, 148.9, 165.7. MS *m*/*z*: 330 (M^+^). HRMS (EI): calcd for C_20_H_14_N_2_O_3_ 330.1004; found 330.1011.

### 3.10. 2-(3-Methylquinolin-2-yl)ethynylbenzaldehyde Oxime (***25b***)

The same procedure as above was carried out with benzaldehyde **24b** (500 mg, 1.84 mmol) to give oxime **25b** (280 mg, 53%) as a red solid. mp 179–181 °C (EtOAc-hexane). IR (ATR) ν = 3051 cm^−1^. ^1^H-NMR (400 MHz, DMSO-*d_6_*) δ 2.65 (s, 3H), 7.50–7.52 (m, 2H), 7.61 (t, *J* = 7.8 Hz, 1H), 7.72–7.76 (t, *J* = 7.8 Hz, 2H), 7.89–7.93 (m, 2H), 8.01 (d, *J* = 7.8 Hz, 1H), 8.29 (s, 1H), 8.64 (s, 1H), 11.71 (s, 1H). ^13^C-NMR (100 MHz, DMSO-*d_6_*) δ 19.8, 89.9, 93.6, 121.0, 125.4, 127.80, 127.83, 128.1, 128.9, 130.0, 130.2, 130.6, 132.9, 133.6, 134.9, 136.1, 143.9, 146.4, 146.8. MS *m*/*z*: 286 (M^+^). HRMS (EI): calcd for C_19_H_14_N_2_O 286.1106; found 286.1115.

### 3.11. 2-(3-Methoxymethylquinolin-2-yl)ethynylbenzaldehyde Oxime (***25c***)

The same procedure as above was carried out with benzaldehyde **24c** (150 mg, 0.50 mmol) to give oxime **25c** (137 mg, 86%) as a red solid. mp 151–153 °C (EtOAc-hexane). IR (ATR) ν = 3055 cm^−1^. ^1^H-NMR (400 MHz, DMSO-*d_6_*) δ 3.47 (s, 3H), 4.79 (s, 2H), 7.48–7.55 (m, 2H), 7.65 (t, *J* = 7.5 Hz, 1H), 7.74–7.76 (m, 1H), 7.80 (t, *J* = 7.5 Hz, 1H), 7.90 (d, *J* = 7.5 Hz, 1H), 8.02–8.05 (m, 2H), 8.41 (s, 1H), 8.66 (s, 1H), 11.72 (s, 1H). ^13^C-NMR (100 MHz, DMSO-*d_6_*) δ 58.2, 71.0, 89.4, 92.1, 120.3, 124.9, 126.9, 127.7, 128.0, 128.4, 129.6, 130.1, 130.2, 132.7, 133.1, 134.5, 134.7, 141.6, 145.9, 146.9. MS *m*/*z*: 316 (M^+^). HRMS (EI): calcd for C_20_H_16_N_2_O_2_ 316.1212; found 316.1216.

### 3.12. 3-(3-Methylquinolin-2-yl)isoquinolin-1-one (***27b***) and 4-Acetoxy-3-(3-methylquinolin-2-yl)isoquinoline (***28b***)

A solution of oxime **25b** (40 mg, 0.14 mmol) in 1,2-dichlorobenzene (4 mL) was stirred at 80 °C for 1 h. After the removal of the solvent, Ac_2_O (4 mL) was added to the residue, and the mixture was stirred at 110 °C for 3 h. After the removal of the solvent, the residue was purified by column chromatography (EtOAc/hexane 1:1 *v*/*v*) to give isoquinolone **27b** (2 mg, 6%) and the 4-acetoxyisoquinoline **28b** (15 mg, 33%).

**27b**: a white solid. mp 214–215 °C (EtOAc-hexane). IR (ATR) ν = 1628 cm^−1^. ^1^H-NMR (400 MHz, CDCl_3_) δ 2.88 (s, 3H), 7.55–7.60 (m, 2H), 7.68–7.76 (m, 3H), 7.80 (d, *J* = 8.2 Hz, 1H), 8.10 (s, 1H), 8.13 (d, *J* = 8.2 Hz, 1H), 8.50 (d, *J* = 8.2 Hz, 1H), 10.64 (br s, 1H). ^13^C-NMR (100 MHz, CDCl_3_) δ 22.5, 108.8, 126.6, 126.7, 127.4, 127.6, 127.7, 127.8, 128.9, 129.1, 129.7, 132.6, 136.2, 137.6, 139.3, 145.7, 148.7, 162.6. MS *m*/*z*: 286 (M^+^). HRMS (EI): calcd for C_19_H_14_N_2_O 286.1106; found 286.1109.

**28b**: an orange oil. IR (ATR) ν = 1770 cm^−1^. ^1^H-NMR (400 MHz, CDCl_3_) δ 2.09 (s, 3H), 2.48 (s, 3H), 7.55 (t, *J* = 7.5 Hz, 1H), 7.65–7.72 (m, 2H), 7.77–7.82 (m, 2H), 7.97 (d, *J* = 7.5 Hz, 1H), 8.08–8.12 (m, 3H), 9.28 (s, 1H). ^13^C-NMR (100 MHz, CDCl_3_) δ 19.4, 20.5, 121.2, 126.8, 126.9, 127.5, 128.0, 128.7, 129.2, 129.5, 131.0, 131.1, 137.0, 140.5, 143.9, 146.1, 149.4, 156.2, 168.7. MS *m*/*z*: 328 (M^+^). HRMS (EI): calcd for C_21_H_16_N_2_O_2_ 328.1212; found 328.1222.

### 3.13. 3-(3-Methoxymethylquinolin-2-yl)isoquinoline N-Oxide (***26c***)

A solution of oxime **25c** (150 mg, 0.47 mmol) in 1,2-dichlorobenzene (10 mL) was stirred at 80 °C for 1 h. After the removal of the solvent, the residue was crystallized from Et_2_O to give *N*-oxide **26c** (109 mg, 73%) as a white solid. mp 222–223 °C (CHCl_3_). ^1^H-NMR (400 MHz, CDCl_3_) δ 3.32 (s, 3H), 4.56 (d, *J* = 12.8 Hz, 1H), 4.84 (d, *J* = 12.8 Hz, 1H), 7.60–7.69 (m, 3H), 7.74 (t, *J* = 8.2 Hz, 1H), 7.79 (d, *J* = 8.2 Hz, 1H), 7.86 (d, *J* = 8.2 Hz, 1H), 7.92 (d, *J* = 8.2 Hz, 1H), 8.01 (s, 1H), 8.15 (d, *J* = 8.2 Hz, 1H), 8.37 (s, 1H), 8.90 (s, 1H). ^13^C-NMR (100 MHz, CDCl_3_) δ 58.6, 71.5, 124.6, 125.7, 127.2, 127.5, 127.6, 128.2, 129.0, 129.4, 129.6 (3C), 129.7, 132.4, 134.9, 136.3, 146.4, 147.2, 151.2. MS *m*/*z*: 316 (M^+^). HRMS (EI): calcd for C_20_H_16_N_2_O_2_ 316.1212; found 316.1222.

### 3.14. 3-(3-Methoxymethylquinolin-2-yl)isoquinolin-1-one (***27c***) and 4-Acetoxy-3-(3-methoxymethylquinolin-2-yl)isoquinoline (***28c***)

A solution of *N*-oxide **26c** (30 mg, 0.095 mmol) in Ac_2_O (2 mL) was heated at 50 °C under microwave irradiation for 3.5 h. After the removal of the solvent, the residue was purified by column chromatography (EtOAc/hexane 1:1 *v*/*v*) to give isoquinolone **27c** (22 mg, 73%) and 4-acetoxyisoquinoline **28c** (7 mg, 21%).

**27c**: a white solid. mp 156–157 °C (EtOAc-hexane). IR (ATR) ν = 1624 m^−1^. ^1^H NMR (400 MHz, CDCl_3_) δ 3.62 (s, 3H), 4.79 (s, 2H), 7.34 (s, 1H), 7.54–7.63 (m, 2H), 7.68–7.73 (m, 2H), 7.79 (t, *J* = 8.2 Hz, 1H), 7.88 (d, *J* = 8.2 Hz, 1H), 8.16 (d, *J* = 8.2 Hz, 1H), 8.35 (s, 1H), 8.49 (d, *J* =8.2 Hz, 1H), 10.50 (br s, 1H). ^13^C-NMR (100 MHz, CDCl_3_) δ 58.2, 72.6, 108.9, 126.7, 127.3, 127.4, 127.5 (2C), 127.7, 127.8, 128.2, 129.2, 130.7, 132.6, 135.9, 137.7, 139.5, 146.7, 150.0, 162.6. MS *m*/*z*: 316 (M^+^). HRMS (EI): calcd for C_20_H_16_N_2_O_2_ 316.1212; found 316.1208.

**28c**: a yellow solid. mp 106–108 °C (EtOAc-hexane). IR (ATR) ν = 1774 cm^−1^. ^1^H-NMR (400 MHz, CDCl_3_) δ 2.13 (s, 3H), 3.41 (s, 3H), 4.73 (s, 2H), 7.58 (t, *J* = 8.2 Hz, 1H), 7.69–7.73 (m, 2H), 7.80 (t, *J* = 8.2 Hz, 1H), 7.90 (d, *J* = 8.2 Hz, 1H), 7.99 (d, *J* = 8.2 Hz, 1H), 8.09 (d, *J* = 8.2 Hz, 1H), 8.13 (d, *J* = 8.2 Hz, 1H), 8.46 (s, 1H), 9.25 (s, 1H). ^13^C-NMR (100 MHz, CDCl_3_) δ 20.6, 58.6, 71.7, 121.4, 126.9, 127.5, 127.6, 127.7, 128.2, 129.1, 129.3, 129.5, 131.1, 131.2, 131.9, 134.6, 140.7, 143.0, 146.5, 149.1, 154.2, 168.7. MS *m*/*z*: 358 (M^+^). HRMS (EI): calcd for C_22_H_18_N_2_O_3_ 358.1317; found 358.1323.

### 3.15. 4-Hydroxy-3-(3-methoxymethylquinolin-2-yl)isoquinoline (***29***)

A solution of 4-acetoxyisoquinoline **28c** (28 mg, 0.078 mmol) in THF (3 mL) was added dropwise to a suspension of LiAlH_4_ (8 mg, 0.20 mmol) in THF (3 mL) under ice cooling, and then stirred at 70 °C for 75 min. After quenching with H_2_O, the reaction mixture was filtrated through a Celite pad and washed with H_2_O and EtOAc. Next, the filtrate was extracted with EtOAc. The organic layer was washed with brine, dried with Na_2_SO_4_, and evaporated in vacuo. The residue was purified by column chromatography (EtOAc/hexane 3:15 *v*/*v*) to give 4-hydroxyquinoline **29** (22 mg, 87%) as an orange solid. mp 139–140 °C (EtOH). IR (ATR) ν = 3055 cm^−1^. ^1^H-NMR (400 MHz, CDCl_3_) δ 1.25 (br s, 1H), 3.66 (s, 3H), 5.47 (s, 2H), 7.56 (t, *J* = 7.8 Hz, 1H), 7.67–7.75 (m, 3H), 7.88 (d, *J* = 7.8 Hz, 1H), 7.94 (d, *J* = 7.8 Hz, 1H), 8.06 (d, *J* = 7.8 Hz, 1H), 8.51 (d, *J* = 7.8 Hz, 1H), 8.66 (s, 1H), 8.81 (s, 1H). ^13^C-NMR (100 MHz, CDCl_3_) δ 58.8, 73.3, 123.0, 125.6, 126.6, 126.7, 126.7, 127.6, 128.8, 128.9, 129.1, 129.6, 130.1, 132.6, 135.6, 139.9, 141.7, 156.2, 156.3. MS *m*/*z*: 316 (M^+^). HRMS (EI): calcd for C_20_H_16_N_2_O_2_ 316.1212; found 316.1216.

### 3.16. Rosettacin (***5***)

We added dropwise conc. H_2_SO_4_ (0.5 mL) under ice cooling to a suspension of isoquinolone **27c** (13 mg, 0.041 mmol) in EtOH (1 mL) and the solution was stirred at 110 °C for 12 h. After cooling to ambient temperature, the solvent was evaporated in vacuo. The residue was diluted with H_2_O (1 mL), and then was alkalified with 1 M aqueous NaOH. The resulting mixture was extracted with CH_2_Cl_2_. The organic layer was washed with brine, dried with Na_2_SO_4_, and evaporated in vacuo. The residue was washed with Et_2_O and filtrated in vacuo to give rosettacin (**5**) (10 mg, 88%) as a yellow solid. mp 292–294 °C (EtOAc-hexane, lit. [26] mp 288 °C). IR (ATR) ν = 1651 cm^−1^. ^1^H-NMR (400 MHz, CDCl_3_) δ 5.39 (s, 2H), 7.57–7.65 (m, 2H), 7.68 (s, 1H), 7.74 (t, *J* = 8.2 Hz, 1H), 7.79–7.83 (m, 2H), 7.92 (d, *J* = 8.2 Hz, 1H), 8.23 (d, *J* = 8.2 Hz, 1H), 8.35 (s, 1H), 8.56 (d, *J* = 8.2 Hz, 1H). ^13^C-NMR (100 MHz, CDCl_3_) δ 49.5, 101.1, 126.1, 127.3 (2C), 127.5 (2C), 128.0 (2C),128.8, 129.5, 130.2, 130.7, 132.5, 137.5, 140.0, 148.9, 153.7, 161.1 MS *m*/*z*: 284 (M^+^). HRMS (EI): calcd for C_19_H_12_N_2_O 284.0950; found 284.0952.

### 3.17. 3-Methoxymethyloxymethyl-2-[(3-methoxymethylquinolin-2-yl)ethynyl]benzaldehyde (***31***)

We added a solution of 2-ethynylbenzaldehyde **30** (82 mg, 0.40 mmol) in THF (1 mL) to a solution of 2-iodoquinoline **23** (100 mg, 0.33 mmol), CuI (3.0 mg, 0.016 mmol), PdCl_2_(PPh_3_)_2_ (23 mg, 0.033 mmol) and Et_3_N (1 mL, 21.63 mmol) in THF (2 mL). The reaction mixture was stirred at 60 °C for 30 min. After cooling to ambient temperature, the reaction mixture was filtrated through Celite pad, washed with EtOAc, and the filtrate was evaporated in vacuo. The residue was purified by column chromatography (EtOAc/hexane 1:9 *v*/*v*) to give 2-alkynylbenzaldehyde **31** (109 mg, 87%) as a yellow solid. mp 87–88 °C (EtOAc-hexane). IR (ATR) ν = 1693 cm^−1^. ^1^H-NMR (400 MHz, CDCl_3_) δ 3.43 (s, 3H), 3.56 (s, 3H), 4.84 (s, 2H), 4.87 (s, 2H), 5.01 (s, 2H), 7.54–7.61 (m, 2H), 7.74 (t, *J* = 7.8 Hz, 1H), 7.85 (d, *J* = 7.8 Hz, 2H), 7.95 (d, *J* = 7.8 Hz, 1H), 8.12 (d, *J* = 7.8 Hz, 1H), 8.28 (s, 1H), 10.77 (s, 1H). ^13^C-NMR (100 MHz, CDCl_3_) δ 55.6, 58.8, 67.1, 71.6, 85.5, 96.4, 98.2, 123.7, 126.6, 127.5, 127.6, 127.8, 129.2, 129.4, 130.0, 132.8, 132.9, 134.4, 136.7, 141.4, 142.1, 147.6, 191.5. MS *m*/*z*: 375 (M^+^). HRMS (EI): calcd for C_23_H_21_NO_4_ 375.1471; found 375.1477.

### 3.18. 3-Methoxymethyloxymethyl-2-[(3-methoxymethylquinolin-2-yl)ethynyl]benzaldehyde Oxime (***32***)

A mixture of benzaldehyde **31** (247 mg, 0.66 mmol), NH_2_OH·HCl (92 mg, 1.32 mmol), and AcONa (108 mg, 1.32 mmol) in EtOH (7 mL) was stirred at rt for 2 h. After removal of solvent, the residue was diluted with H_2_O, and then filtrated off to give crude oxime **32** (208 mg, 81%) as a red solid. The product was recrystallized from EtOAc-hexane. mp 130–131 °C (EtOAc-hexane). IR (ATR) ν = 3734 cm^−1^. ^1^H-NMR (400 MHz, DMSO-*d_6_*) δ 3.35 (s, 3H), 3.47 (s, 3H), 4.77 (s, 2H), 4.83 (s, 2H), 4.87 (s, 2H), 7.51–7.55 (m, 1H), 7.59 (d, *J* = 7.8 Hz, 1H), 7.66 (t, *J* = 7.8 Hz, 1H), 7.79–7.86 (m, 2H), 8.04–8.06 (m, 2H), 8.44 (s, 1H), 8.70 (s, 1H), 11.71 (s, 1H). ^13^C-NMR (100 MHz, DMSO-*d_6_*) δ 54.9, 58.1, 67.0, 70.9, 86.6, 95.8, 96.8, 118.9, 123.8, 127.0, 127.8, 128.0, 128.4, 129.8, 130.2, 132.7, 134.6, 135.0, 141.3, 141.5, 146.0, 146.9. MS *m*/*z*: 390 (M^+^). HRMS (EI): calcd for C_23_H_22_N_2_O_4_ 390.1580; found 390.1590.

### 3.19. 5-Methoxymethyloxymethyl-3-(3-methoxymethylquinolin-2-yl)isoquinoline N-Oxide (***33***)

A solution of oxime **32** (60 mg, 0.15 mmol) in 1,2-dichlorobenzene (4 mL) was stirred at 180 °C for 12 h. After removal of solvent, the residue was purified by column chromatography (MeOH) to give *N*-oxide **33** (21 mg, 36%). The obtained **33** contained a small amount of impurity and could not be further purified.

### 3.20. 5-Methoxymethyloxymethyl-3-(3-methoxymethylquinolin-2-yl)isoquinolin-1-one (***34***) and 4-Acetoxy-5-methoxymethyloxymethyl-3-(3-methoxymethylquinolin-2-yl)isoquinoline (***35***)

A solution of *N*-oxide **33** (15 mg, 0.038 mmol) in Ac_2_O (1 mL) was heated at 50 °C under microwave irradiation for 24 h. After removal of the solvent, the residue was purified by column chromatography (EtOAc/hexane 1:1 *v*/*v*) to give isoquinolone **34** (8 mg, 54%) and the 4-acetoxyisoquinoline **35** (3 mg, 20%).

**34**: a yellow solid. mp 113–114 °C (EtOAc-hexane). IR (ATR) ν = 1628 cm^−1^. ^1^H-NMR (400 MHz, CDCl_3_) δ 3.42 (s, 3H), 3.62 (s, 3H), 4.77 (s, 2H), 4.82 (s, 2H), 4.97 (s, 2H), 7.54 (t, *J* = 7.8 Hz, 1H), 7.62 (t, *J* = 7.8 Hz, 1H), 7.67 (s, 1H), 7.75–7.82 (m, 2H), 7.89 (d, *J* = 7.8 Hz, 1H), 8.17 (d, *J* = 7.8 Hz, 1H), 8.36 (s, 1H), 8.49 (d, *J* = 7.8 Hz, 1H), 10.61 (s, 1H). ^13^C-NMR (100 MHz, CDCl_3_) δ 29.7, 55.5, 58.4, 66.9, 72.7, 95.7, 105.4, 127.2, 127.3, 127.4, 127.7, 127.8, 128.3, 129.3, 130.7, 133.2, 133.8, 136.1, 136.5, 139.6, 146.8, 150.1, 162.8. MS *m*/*z*: 390 (M^+^). HRMS (EI): calcd for C_23_H_22_N_2_O_4_ 390.1580; found 390.1574.

**35**: an orange oil. IR (ATR) ν = 1770 cm^−1^. ^1^H-NMR (400 MHz, CDCl_3_) δ 1.95 (s, 3H), 3.35 (s, 3H), 3.44 (s, 3H), 4.63 (s, 2H), 4.71 (s, 2H), 5.19 (s, 2H), 7.58 (t, *J* = 7.8 Hz, 1H), 7.65 (d, *J* = 7.8 Hz, 1H), 7.71 (t, *J* = 7.8 Hz, 1H), 7.87 (d, *J* = 7.8 Hz, 1H), 7.90 (d, *J* = 7.8 Hz, 1H), 8.04 (d, *J* = 7.8 Hz, 1H), 8.13 (d, *J* = 7.8 Hz, 1H), 8.48 (s, 1H), 9.23 (s, 1H). ^13^C NMR (100 MHz, CDCl_3_) δ 21.0, 55.4, 58.7, 67.8, 70.9, 94.8, 126.9, 127.66, 127.71, 127.8, 128.1, 129.0, 129.2, 129.9, 130.9, 132.3, 132.4, 132.6, 134.4, 140.7, 145.0, 146.2, 149.8, 154.4, 168.7. MS *m*/*z*: 432 (M^+^). HRMS (EI): calcd for C_25_H_24_N_2_O_5_ 432.1685; found 432.1694.

### 3.21. Acuminatine (***7***)

We added dropwise conc. H_2_SO_4_ (2 mL) under ice cooling to a suspension of isoquinolone **34a** (30 mg, 0.077 mmol) in EtOH (4 mL) and then stirred the mixture at 110 °C for 16 h. After cooling to an ambient temperature, the solvent was evaporated in vacuo. The residue was diluted with H_2_O (2 mL) and then was alkalified with 1 M aqueous NaOH. The resulting mixture was extracted with CH_2_Cl_2_. The organic layer was washed with water and brine, dried with Na_2_SO_4_, and evaporated in vacuo. The residue was purified by column chromatography (EtOAc/hexane 4:6 *v*/*v*) to give acuminatine (**7**) (18 mg, 79%). mp 290–292 °C (EtOAc-hexane). IR (ATR) ν = 1770 cm^−1^. ^1^H-NMR (400 MHz, CDCl_3_) δ 2.74 (s, 3H), 5.38 (s, 2H), 7.47 (t, *J* = 7.8 Hz, 1H), 7.58 (d, *J* = 7.8 Hz, 1H), 7.62 (t, *J* = 7.8 Hz, 1H), 7.79–7.83 (m, 2H), 7.91 (d, *J* = 7.8 Hz, 1H), 8.24 (d, *J* = 8.7 Hz, 1H), 8.35 (s, 1H), 8.43 (d, *J* = 7.8 Hz, 1H). ^13^C-NMR (100 MHz, CDCl_3_) δ 19.6, 49.4, 98.1, 125.5, 126.4, 127.1, 127.3, 128.0, 128.1, 128.9, 129.4, 130.3, 130.9, 133.3, 135.0, 136.6, 139.6, 148.9, 153.9, 161.3. MS *m*/*z*: 298 (M^+^). HRMS (EI): calcd for C_20_H_14_N_2_O 298.1106; found 298.1111.

## 4. Conclusions

We developed a synthetic route to create the pentacyclic scaffold of the aromathecin family by constructing the indolizidine moiety after isoquinolone synthesis. The key step of our isoquinolone synthesis employed a thermal cyclization of 2-alkynylbenzaldehyde oxime to the isoquinoline *N*-oxide, followed by the performance of a Reissert–Henze-type reaction. Under the optimum reaction conditions for the Reissert–Henze-type reaction step, microwave irradiation-assisted heating of the purified *N*-oxide in Ac_2_O at 50 °C reduced the formation of the 4-acetoxyisoquinoline byproduct to deliver the desired isoquinolone at a moderate yield. The employed eight-step sequence afforded rosettacin at a 23.8% overall yield. Furthermore, the synthesis of acuminatine was achieved by applying the developed strategy. Future research will focus on the use of this strategy to efficiently synthesize a variety of polycyclic compounds, including alkaloids with promising therapeutic profiles.

## Data Availability

The data presented in this study are available in Appendix A.

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
