# Peer review of "Novel Approach to the Construction of Fused Indolizine Scaffolds: Synthesis of Rosettacin and the Aromathecin Family of Compounds"

_molecules, 2023, doi:10.3390/molecules28104059_

Round 1
Reviewer 1 Report
Aromathecin-like compounds are a kind of new fused indolizine scaffolds with pretty good anticancer activity, this manuscript reported a new total synthetic process for the aromathecin derivatives based on the retrosynthetic analysis.
The authors’ group has done great deal of works on the synthesis of similar scaffolds, the reported process is also a versatile route for preparing aromathecin derivatives, and many of starting materials have been tried with good yields in this manuscript. It should be interested and helpful to the readers, so I suggest publishing this paper after minor revised.
1. Introduction, it is better to represent the synthetic methods of the aromathecin derivatives reported in the literatures.
2. In Scheme 5, the structure of 29 should be presented.
Reviewer 2 Report
1) Should cite the appropriate references of the Figure 1 data.
2) Authors should include more biological information on rosettacin and aromathecin derivatives in the introduction section.
3) The structure of the “compound 28c” acetyl group is in a different position in ‘scheme 5’ and ‘figure 2’.
4) Based on the NOESY and HMBC correlation data, how does the acetyl group exist in the 4th position? Explain.
5) In scheme 6, “compound 5” yields 88% or 23.8%.
The manuscript described the synthesis of rosettacin and aromathecin analogs under microwave irradiation conditions. Systematically explained reaction protocol during the synthesis of fused indolizine scaffolds and reported the different characterization techniques used to confirm the synthesized molecules. Further, NOESY and HMBC correlation data were provided for structural isomers' confirmation. Therefore, this protocol is useful to develop for synthesising anticancer drug candidates; hence, after minor modifications, this could be interesting for readers and may be published in the journal.
Reviewer 3 Report
The paper deals with the development of synthetic strategies for the preparation of aromathecine-based compounds, which, containing the same central indazolidine structure as the camptothecin derivatives, could also be endowed with antitumor activity.
The key step of the synthetic route, which consists of eight steps, involves the thermal cyclization of 2-alkynylbenzaldehyde oxime to isoquinoline N-oxide, followed by a Reissert-Henze-type reaction. The work is a continuation of previous research by the authors (10 references by the same authors confirm this). The paper falls within the scope of the journal, the achieved results are interesting and worthy to be published. Anyway, the manuscript requires small changes before it can be accepted.
Here are some suggestions:
- There are some instances of language discrepancies and typographical errors in the manuscript (for example in Figure 1 legend, “scafford” should be corrected in “scaffold”, and so on). The authors are requested to get the manuscript critically checked by an English-speaking reviewer.
- The type of the paper should be specified by the authors (Article, Review, Communication, etc.).
- Keywords should not contain the same words present in the title (i.e: romathecin, rosettacin).
- In the experimental section, the authors should standardize the units of measurement: always g or always mg, always mol or always mmol, and establish the number of digits after the decimal point.
- a double check of the NMR spectra is required.
- The conclusions should be extended to provide a more comprehensive summary of the entire work.
- In “Funding: Please add: This research received no external funding”. Remove “Please add:”
Bibliographic references:
Please add after reference 8: “Expert Opinion on Investigational Drugs, 2009, 18(5), pp. 555–568” ;
Ref 15: the end stop is missing;
Ref 31: "Marco" is the first name of the author. Please check all authors’ names.
After these modifications, the paper could be accepted.
There are some instances of language discrepancies and typographical errors in the manuscript. The authors are requested to get the manuscript critically checked by an English-speaking reviewer.
